# Relationship between Recovery from COVID-19-Induced Smell Loss and General and Oral Health Factors

**DOI:** 10.3390/medicina58020283

**Published:** 2022-02-14

**Authors:** Georgia Catton, Alexander Gardner

**Affiliations:** Department of Restorative Dentistry, Dundee Dental Hospital and School, University of Dundee, Dundee DD1 4HR, UK; g.s.catton@dundee.ac.uk

**Keywords:** COVID-19, anosmia, oral health, dental care

## Abstract

*Background and Objectives*: Loss of smell is one of the strongest predictors of coronavirus disease 2019 (COVID-19) and can persist long after other symptoms have resolved. “Long” cases (>28 days) of smell dysfunction present future challenges to medical and dental professionals, as there is a lack of evidence on the causes and any exacerbating or relieving factors. This study aimed to explore the persistence of COVID-19-induced smell loss and association with physical, lifestyle and oral health factors. *Materials and Methods*: This study was a cross-sectional survey of 235 participants. Recovery of smell was explored, comparing rapid recovery (≤28 days) with prolonged recovery (>28 days). Associative factors included age, sex, illness severity, diet, BMI, vitamin D supplementation, antidepressants, alcohol use, smoking, brushing frequency, flossing, missing teeth, appliances and number of dental restorations. *Results*: Smell loss showed 87% resolution within 30 days. Prolonged smell loss was significantly associated with older age (mean ± 95%, CI = 31.53 ± 1.36 years for rapid recovery vs. mean ± 95%, CI = 36.0 ± 3 years for prolonged recovery, *p* = 0.003) and increased self-reported illness severity (mean ± 95%, CI = 4.39 ± 0.27 for rapid recovery vs. 5.01 ± 0.54 for prolonged recovery, *p* = 0.016). Fisher’s exact test revealed flossing was associated with rapid recovery, with flossers comprising 75% of the rapid-recovery group, compared to 56% in the prolonged-recovery group (odds ratio ± 95%, CI = 2.26 (1.23–4.15), *p* = 0.01). All other factors were not significantly associated (*p* > 0.05). *Conclusions*: Increased age and illness severity were associated with prolonged smell recovery. Use of floss was the only modifiable factor associated with rapid recovery of smell loss. As 87% of cases resolve within 30 days, future studies may benefit from targeted recruitment of individuals experiencing prolonged sense loss. This would increase statistical confidence when declaring no association with the other factors assessed, avoiding type II errors.

## 1. Introduction

Alteration and loss of sense of taste and smell was widely observed in patients early in the severe acute respiratory syndrome coronavirus 2 (SARS-CoV-2) pandemic and quickly adopted by national and international bodies as a defining symptom of the disease, alongside fever and cough [1,2]. Indeed, sensory symptoms have been shown as one of the best predictors of the disease, short of the current gold-standard polymerase chain reaction (PCR) test [3,4]. A recent meta-analysis suggests sensory symptoms display comparable negative predictive value (0.78 vs. 0.80) and lower positive predictive value (0.78 vs. 0.98) when compared to PCR testing [3,5].

While there are relatively abundant reports of the prevalence of sensory symptoms in COVID-19, there are comparatively few studies on recovery from these symptoms [6,7]. In recent months, primary care dental services have been remobilising as the United Kingdom (UK) emerges from the worst of the pandemic, with the profession facing several challenges. Amongst these challenges, there is a likelihood of increased cases of patients reporting “long” COVID-19 symptoms, defined as symptoms persisting beyond 4 weeks [8]. Sensory complaints in particular have been shown to persist beyond other symptoms of the disease [9]. An exact mechanism for the sensory symptoms caused by COVID-19 infection has not yet been elucidated; however, inflammation is believed to play a key role, either peripherally at taste and smell receptors or centrally within the central nervous system (CNS) [10]. This process involves viral entry mediated by binding of viral surface protein via angiotensin-converting enzyme 2 (ACE-2) expressed on oral and nasal tissues. The virus may enter and destroy chemosensory cells directly or damage nerve cells involved in processing chemosensory signals [11]. Indirect damage may occur by locally or systemically raised inflammatory markers, such as interleukins and Tumour necrosis factor alpha (TNF-α). A further proposed mechanism is the presence of central hypoxia impairing sensory function, although there is less evidence for this hypothesis [12]. Interestingly, unlike with other respiratory viruses, smell loss appears to be independent of any nasal congestion [13].

The present study is a cross-sectional survey of individuals who have suffered COVID-19 and focuses on the recovery of smell in association with factors such as age; sex; smoking; alcohol use; and diet and dental factors, such as frequency of brushing, flossing, number of missing teeth, restorations and wearing appliances. The survey was hosted on the news-aggregator website reddit, in the community reddit.com/r/COVID19positive. Such an approach allows for crowdsourcing of responses, with over 170 studies relating to COVID-19 having been hosted at time of writing [14]. The rationale for selecting these variables stemmed from early observations from the medical community in the first few months of the pandemic of factors associated with COVID-19 infection. Many studies focused on outcomes such as hospitalisation, need for ventilation or mortality, with smell loss being considerably less studied. Other factors were chosen based on pre-existing associations with smell function, predating the COVID-19 pandemic. Age was observed as major risk factor for disease severity and mortality, and males were also observed to experience more severe disease early in the pandemic [15,16]. Diet and obesity are major risk factors for COVID-19 severity [17]; hence, fruit and vegetable intake and body mass index (BMI) were included in this study. Similarly, vitamin D deficiency was implicated in disease severity early in the pandemic, as well as in smell function [18,19]. Alcohol consumption was also recognised as an early risk factor for severe disease [20], as was smoking [21], which is a long recognised adverse health behaviour. Smoking is also known to diminish smell function in the absence of other disease [22].

Considerably less was known about the relation between oral health and COVID-19 in the early stages of the pandemic. The rationale for exploring oral hygiene, particularly flossing, was due to the observations that periodontal disease is associated with localised and systemic inflammation, which can impair immune function. Periodontal pockets may also harbour COVID-19 virions, representing a mechanism of entry via the oral cavity [11]. Furthermore, assessment of missing teeth not due to trauma or orthodontics is an indicator of past periodontal disease, and a recent study found 70% of COVID-19 patients had lost teeth due to severe periodontal disease [11]. The evaluation of dental restorations and oral appliances was related to the role of oral biofilms in modulating taste and smell function via retronasal olfaction [23,24]. Factors such as appliance wearing, tooth brushing habits and the restorative state of the dentition are recognised as modifying the oral microbiome, which, in turn, may alter oral function [25]. While less studied than the oral microbiome, the nasal microbiome may similarly have an impact of the process of olfaction [26], and its composition has been implicated in differentiating COVID-19 patients pre- and post-recovery [27].

The aim of this work was to investigate associations between self-reported smell loss in COVID-19 and biometric, lifestyle and oral health factors. The rationale for exploring these factors was to add to the literature base, particularly for medical and dental professionals who may encounter patients experiencing delayed smell recovery. Such cases are observed to occur with increasing frequency both in the authors’ clinical practice and the wider literature [28]. Findings from this work are also intended to provide exploratory pilot data to serve as an indicator for future research directions related to smell loss in COVID-19, primarily by providing a baseline for sample size and split based on the distribution pattern of smell recovery.

## 2. Materials and Methods

### 2.1. Ethical Approval

This study was approved by the University of Dundee School of Health Sciences and Dentistry ethics committee (ref: UOD\SDEN\STAFF\2020\017, 25 June 2020).

### 2.2. Survey Design and Dissemination

A survey was designed using JISC Online Surveys (JISC, Bristol, UK) and hosted on https://www.reddit.com/r/COVID19positive (accessed on 10 March 2021), following written approval from the site administration team. The survey was open to individuals 18 years old or above who had suffered COVID-19 infection between March and August 2020, diagnosed by PCR test, a medical professional or self-diagnosis, where symptoms included acute changes to smell and/or taste. The survey was closed to respondents in October 2020. A breakdown of the variables explored is described below, and the full survey can be found in Appendix A.

### 2.3. Sample Size and Statistical Power

Based on a 95% confidence level, *p* = 0.05 and a precision of ±5%, a sample size of 400 was deemed necessary as a representation of the study population [29]. Preliminary exploration of early responders experiencing sensory loss suggested a cut-off of ~30 days would yield a sampling ratio of between 4:1 and 5:1 (recovery ≤ 28 days: recovery > 28 days). This distribution ratio was sufficient to detect effect sizes differing by 15–20% between groups with alpha = 0.05 and power = 0.8, based on a sample distribution of 234 (195:39) participants. 

### 2.4. Data Gathering and Processing

Survey responses were manually processed and coded. Continuous variables were age, BMI, fruit and vegetable intake and duration of smell loss. Ordinal variables were severity of smell loss, severity of overall illness and severity of congestion, which were defined on an 11-point (0–10) scale, with 0 being no effect and 10 being total loss of sensation. Further ordinal variables were brushing frequency (daily, twice daily, >twice daily), alcohol consumption frequency (never, monthly, weekly, daily), smoking (never, former, current) and number of fillings (0, 1–5, 5–10, >10). Binary variables were biological sex, vitamin D supplementation, antidepressant use, use of floss/interproximal cleaning, missing teeth (not lost due to trauma, orthodontics or impaction) and appliance wearing. Low numbers of African and South Asian respondents prevented statistical analysis of ethnicity.

### 2.5. Statistical Analyses

Analyses and data visualisation were performed in Python3 (Centrum Wiskunde & Informatica, Amsterdam, The Netherlands) and SPSS 27 (IBM, Armonk, NY, USA). Variables were inspected and tested for normality (QQ plots and Shapiro–Wilk test). The relationship between disease severity and recovery times was explored by Spearman’s rank correlation. A step plot for the duration of recovery for smell loss was created. Differences between rapid recovery (≤28 days) and prolonged recovery (>28 days) were then explored. Means were compared by two-tailed *t*-test or Mann–Whitney test where appropriate. Binary variables were compared using Fisher’s exact test, and categorical variables were compared by Pearson’s chi-squared test.

## 3. Results

### 3.1. Resolution of Smell Loss

A total of 421 participants reported some degree of smell, with 235 reporting either full recovery within 28 days or persisting symptoms beyond this time. A total of 207 participants reported full recovery at the time of completing the survey, allowing for the creation of a recovery curve (Figure 1). The recovery curve indicates that within 14 days, 64% of cases had resolved, and within 30 days, 87% had resolved, rising to 96% resolution within 90 days, whereas only two cases (<1%) persisted beyond 120 days. 

### 3.2. Differences between Rapid (≤28 Days) and Prolonged (>28 Days) Recovery

In total, 235 individuals were analysed with regards to smell loss, composed of 173 individuals who reported resolution in under 28 days and 28 individuals who reported smell loss persisting beyond 30 days. A full breakdown of the statistical analyses is presented in Table 1. The main significant findings were that individuals with prolonged recovery were, on average, older than individuals in the rapid-recovery group. Mean and 95% CI age for rapid smell recovery was 31.53 (30.17–32.88) years, compared to 36.0 (33.0–39.0) for prolonged recovery, *p* = 0.003. Data are presented graphically in Figure 2a.

A further significant difference was found for severity of illness when comparing smell-loss recovery. Severity was 4.39 (4.11–4.66) in the rapid-recovery and 5.01 (4.58–5.55) in the prolonged-recovery group, *p* = 0.016. Data are presented in Figure 2b. Interestingly, severity of smell loss itself did not differ between the recovery groups. Smell loss severity was 8.47 (8.12–8.83) vs. 8.58 (7.93–9.24), *p* = 0.44, for rapid vs. prolonged smell recovery.

Of the lifestyle and oral-health parameters analysed, the only significant difference was for the use of floss. A higher proportion of flossers was found in the rapid-recovery groups compared to the prolonged-recovery group. The observed distribution was 129/173 (75%) in the rapid smell recovery compared to 35/62 (56%) in the prolonged smell recovery, odds ratio = 2.26 (1.23–4.15), *p* = 0.01. Flossing and age were found to be statistically independent when comparing mean age of flossers and non-flossers by *t*-test, *p* = 0.68. As shown in Table 1, distributions for the other analyses were generally very close to the expected values, with no significant differences between rapid and prolonged recovery.

## 4. Discussion

### 4.1. Pattern of Smell Recovery

The present study aimed to investigate factors that may be linked to the delayed recovery of smell due to COVID-19. We observed recovery of smell at 14 days to be 64%, which is slightly lower than some studies in Korean and Chinese populations, reporting 75% and 80% recovery at two weeks, respectively [30,31]. It has been reported, however, that East Asian populations experience reduced incidence of sensory dysfunction compared to Caucasian populations [32], although whether this also applies to recovery duration is unknown. In contrast, a study of Italian participants found recovery of both senses at two weeks to be as low as 50% [33]. We did observe 95% recovery of smell at two months, whereas others report recovery of 80–88% [1,34]. Furthermore, there are reports of chemosensory dysfunction at six months in 14% of individuals, a considerably higher proportion than was observed in our sample (<1%) [3]. 

### 4.2. Differences between Rapid and Prolonged Recovery

Our findings suggest that individuals reporting prolonged sensory recovery were significantly older and reported a more severe overall disease than those reporting rapid recovery. Such an association with age has been reported by others, with individuals >40 years reporting slower recovery of smell than those <40 years old [35]. Other work has found age is not a prognostic factor for smell recovery, and literature is sparse with respect to taste recovery [36,37]. Literature is similarly divided on the relationship between sensory recovery and severity of both disease and individual sensory loss. The present study found prolonged smell recovery was associated with significantly higher overall disease severity, although no difference was found for severity of smell loss. Other studies do not report a relationship between disease severity and recovery [31], whereas there are mixed reports associating severity of chemosensory loss with recovery time [1,34,38]. Interestingly, nasal congestion has been positively associated with smell recovery, a finding that the present study did not replicate [35]. 

To our knowledge, there are no reports to date investigating the relationship between oral-health behaviour and lifestyle and sensory recovery post-COVID-19. The only significant difference was found for flossing, which was more prevalent in the rapid-recovery group. The presence of periodontal disease has been previously associated with reduced COVID-19 outcomes, although sensory-dysfunction recovery was not specifically assessed [39,40]. Potential explanations for this include a baseline level of inflammation in periodontitis patients, with periodontally compromised COVID-19 patients having higher levels of C-reactive protein (CRP) [39]. Further evidence has linked CRP with adverse oral health and increased COVID-19 severity [41]. Other inflammatory markers, such as interleukin-6, have been shown to be raised in periodontal disease [41], and interleukin-6 has been shown to be inversely linked with sensory recovery [42]. The present study did not clinically examine periodontal disease, for example, by quantifying pocket depth and bleeding indices. Daily flossing has been shown to significantly reduce clinically measured gingival inflammation and thus bleeding [43,44], indicating daily flossing is associated with greater gingival health and reduced inflammation. Mechanistically, this observation ties in with previous hypotheses for local mechanisms of COVID-19 infection, such as viral infiltration of locally inflamed periodontal pockets [11] or increased systemic inflammatory markers due to prior periodontal infection [39]. Furthermore, many periodontal pathogens, such as *Porphyromonas Gingival* and *Treponema Denticola*, have been implicated in systemic inflammatory disease in organs such as the liver, heart and joints [45]. Increased periodontal pathogens have also been associated with neurological degeneration, particularly smell, in Parkinson’s disease [46]. The management of periodontal pathogens via antiseptic mouthwashes is also a recognised clinical approach [47] and has been demonstrated to have efficacy for COVID-19 viral-load reduction [48]. The present work did not ask about mouthwash use, however, as alteration of taste perception can be induced by chlorhexidine itself [49]. 

### 4.3. Limitations

There are several limitations in the present study. The self-reporting nature of symptoms may result in under-recognition of the symptoms when compared to objective measures of smell loss [50]. Nevertheless, successful research of sensory loss has been conducted by similar means [51,52,53]. The demographics of our sample were imbalanced with regard to ethnicity, precluding statistical analysis. This was largely due to lower numbers of African and South Asian participants. As adverse COVID-19 outcomes have been reported for both groups, exploration of sensory symptoms in these groups is a key area for future research [54,55]. As most participants did not report prolonged sensory loss, statistical power in our sample distribution could only reliably detect relatively large effect sizes due to the imbalance between early and prolonged recovery. This is true of all datasets in this area of research. Thus, type II errors may be increased in our sample; however, based on our observed distributions for most variables, any statistically significant effect size would be unlikely to translate into clinical significance. A further limitation lies in the inclusion of cases diagnosed by a doctor or self-diagnosed based on symptoms. Further analyses of PCR tests confirmed cases do yield the same pattern of significant results at the α = 0.05 level (Appendix A). Given the nature of inequitable access to testing at the stage of the pandemic when this survey was conducted, particularly in the USA, where most participants were from, the decision to include those unable to access a test was made [56]. Finally, we recommend cautious interpretation of the association of flossing and sensory recovery, which, of course, cannot be interpreted as causative. For example, flossing may be an inadvertent indicator of a generally more health-focused individual or perhaps someone more compliant and receptive to healthcare advice. Despite a statistically significant difference in flossers between smell-recovery groups (75% of rapid smell recovery vs. 58% of prolonged smell recovery) the clinical significance cannot be automatically assumed. 

## 5. Conclusions

The present work found that smell was recovered following COVID-19 within 30 days in 87% of cases. Age and disease severity were positively associated, and use of floss was negatively associated with prolonged recovery (>28 days) of smell. Non-significant factors for sensory recovery included sex, diet, BMI, vitamin D, antidepressants, alcohol use, smoking, brushing frequency, missing teeth, appliances and number of restorations. Several previous studies have related periodontal disease with adverse COVID-19 outcomes, including ventilator use and mortality. The present work adds to the literature by suggesting recovery of smell loss may also be associated with flossing, supporting previous hypotheses of viral entry via inflamed gingival sulci. The other findings linking smell recovery with age and illness severity and the lack of association between the other factors studied adds to the collective understanding of COVID-19. Within the limitations of this study, further research and studies with higher participant numbers are needed to increase confidence in the lack of associations found to minimise the risk of type II errors. Future work will benefit from targeted recruitment of patients experiencing prolonged symptoms; our study suggests these comprise 13% of patients. Focus on these patients may also be highly useful in further elucidating the mechanism of COVID-19-induced smell dysfunction, for example, by exploring their oronasal tissue expression of ACE-2 or their oral and nasal microbiomes. 

## Figures and Tables

**Figure 1 medicina-58-00283-f001:**
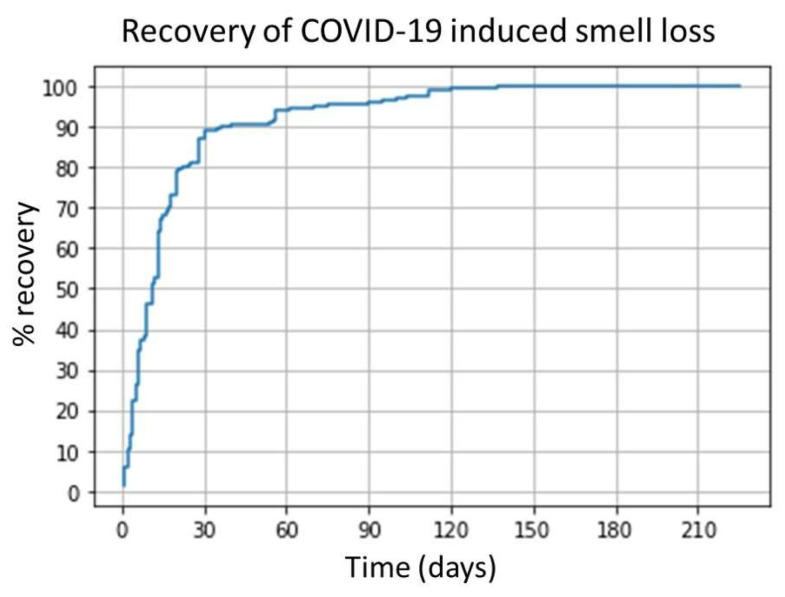
Recovery curve for smell loss, *n* = 207.

**Figure 2 medicina-58-00283-f002:**
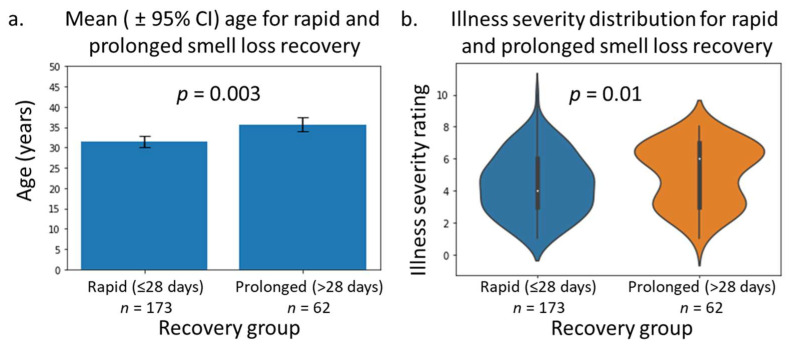
(**a**) Comparison of age between rapid and prolonged smell recovery; *p*-values are for two-tailed *t*-test. (**b**) Comparison of illness severity between rapid and prolonged smell recovery; *p*-values are for Mann–Whitney test.

**Table 1 medicina-58-00283-t001:** Comparison of short (≤28 days) and long (>28 days) smell loss.

Variable	Test	Short Smell Loss (≤28 Days) Mean ± 95% CI		Long Smell Loss (>28 Days) Mean ± 95% CI		Test Statistic	*p*-Value
Age	*t*-test	31.53 (30.17–32.88)		36.0 (33.0–39.0)		−3.04	0.003 **
BMI	*t*-test	26.81 (25.95–27.69)		25.92 (24.64–27.20)		1.077	0.28
Fruit/vegetable intake	*t*-test	2.91 (2.66–3.16)		3.07 (2.62–3.52)		−0.608	0.54
Illness severity	Mann–Whitney	4.39 (4.11–4.66)		5.01 (4.58–5.55)		−2.42	0.016 *
Smell-loss severity	Mann–Whitney	8.47 (8.12–8.83)		8.58 (7.93–9.24)		−0.77	0.44
Congestion	Mann–Whitney	3.08 (2.64–3.53)		2.72 (1.92–3.52)		−0.10	0.32
Sex	Fisher’s exact	M	64 (65)	M	24 (23)	NA	0.88
		F	109 (108)	F	38 (39)		
Vitamin D	Fisher’s exact	No	149 (151)	No	56 (54)	NA	0.27
		Yes	24 (22)	Yes	6 (8)		
Antidepressant	Fisher’s exact	No	155 (155)	No	56 (56)	NA	0.54
		Yes	18 (18)	Yes	6 (6)		
Flossing	Fisher’s exact	No	44 (52)	No	27 (19)	NA	0.01 **
		Yes	129 (121)	Yes	35 (43)		
Missing teeth	Fisher’s exact	No	145 (145)	No	53 (53)	NA	0.83
		Yes	23 (23)	Yes	9 (9)		
Appliances	Fisher’s exact	No	138 (140)	No	52 (50)	NA	0.57
		Yes	35 (33)	Yes	10 (12)		
Brushing freq.	Pearson’s Chi^2^	Daily	40 (34)	Daily	32 (38)	3.88	0.14
		2 × day	66 (69)	>2 × day	80 (77)		
		>2 × day	5 (8)	>2 × day	11 (8)		
Smoking	Pearson’s Chi^2^	Never	136 (135)	Never	47 (48)	0.21	0.90
		Former	12 (12)	Former	5 (5)		
		Current	25 (26)	Current	10 (9)		
Alcohol freq.	Pearson’s Chi^2^	Never	32 (37)	Never	18 (13)	4.61	0.20
		Monthly	43 (38)	Monthly	9 (14)		
		Weekly	63 (63)	Weekly	23 (23)		
		Daily	35 (35)	Daily	12 (12)		
No. fillings	Pearson’s Chi^2^	None	49 (50)	None	19 (18)	1.99	0.57
		<5	74 (71)	<5	23 (26)		
		5–10	36 (39)	5–10	17 (14)		
		>10	14 (13)	>10	3 (5)		

Fisher’s exact test and Chi^2^ test observed values are shown, followed by expected values in brackets. * = significant at *p* < 0.05, ** = significant at *p* < 0.01.

## Data Availability

The authors will make data available upon reasonable request.

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
