# Peer review of "Relationship between Recovery from COVID-19-Induced Smell Loss and General and Oral Health Factors"

_medicina, 2022, doi:10.3390/medicina58020283_

Round 1

Reviewer 1 Report

The article has improved significantly and this reviewer considers the paper acceptable for publication

Reviewer 2 Report

The paper presents an interesting point of view on a quite neglected area of interest for the broad healthcare universe, making it a significant novel contribution to the actual state-of-the art. Overall, the paper does not present any serious flaws and actually delivers all what is outlined in the Abstract in a frank, unbiased manner.

The overall interest in the paper could be somewhat limited by the fact that the topic faced represents a true niche in the healthcare universe, but this should not prevent this paper from receiving the attention it deserves.

I do not have particular concerns to the publication of this manuscript, therefore I suggest its acceptance after a thorough proofreading for typos throughout the text.

This manuscript is a resubmission of an earlier submission. The following is a list of the peer review reports and author responses from that submission.

Round 1

Reviewer 1 Report

The article entitled “Relationship between recovery of COVID-19 induced smell loss and general and oral health factors” aimed to investigate the persistence of smell loss and associated oral factors after COVID-19 infection.

The topic of the paper is interesting and in line with the journal aim, but the manuscript is not structured properly.  
I ask the authors to introduce and analyze some key concepts necessary to make this manuscript publishable for the journal.
1) The authors must indicate the real aim of the study, the sentence "The aim of this work was to provide evidence for dental professionals to aid in the reassurances and advice they can provide patients who may be experiencing or have experienced prolonged loss of smell following COVID-19" is not clear, the aim of the paper is to evaluate the recovery of the sense of smell in covid positive patients and relate the data to the oral conditions of the patients? Please explain this concept further.
2) the abstract is confusing, please rewrite the entire paragraph.
3) the introduction is very short, several studies have been reported analyzing the oral manifestations of COVID-19, mention and comment on them ( see and discuss doi:10.3390/jcm9103218), it is necessary to stress how chemosensory impairments have been established as a specific indicator of COVID19.
4) What are the data related to the oral condition of the patient? the sentence "The present study did not examine periodontal disease directly, however flossing may be associated with greater gingival health and reduced inflammation" is not consistent, please introduced other data to discuss and analyze previous findings regarding the importance of periodontal disease and how periodontal bacteria affect the oral cavity of the patient (please, see and consider: doi:10.3390/jcm9010284)

Minor issues:

  • Conclusions cannot be reduced to a list of sentences: you must improve them highlighting the limits and the future insights pointed out from this article.
  • The references must be reformatted according to the instructions for the authors (see Journal Articles: Author 1, A.B.; Author 2, C.D. Title of the article. Abbreviated Journal NameYearVolume, page range.)

According to this Reviewer’s consideration, novelty and quality of the paper, publication of the present manuscript is recommended after major revision.

Author Response

Dear Colleague,

Thank you for taking the time to review our manuscript and offer your insight into improving the overall manuscript. Please find out responses to your individual comments detailed below.

The topic of the paper is interesting and in line with the journal aim, but the manuscript is not structured properly.  
I ask the authors to introduce and analyze some key concepts necessary to make this manuscript publishable for the journal.
1) The authors must indicate the real aim of the study, the sentence "The aim of this work was to provide evidence for dental professionals to aid in the reassurances and advice they can provide patients who may be experiencing or have experienced prolonged loss of smell following COVID-19" is not clear, the aim of the paper is to evaluate the recovery of the sense of smell in covid positive patients and relate the data to the oral conditions of the patients? Please explain this concept further.

We have re-written the aims paragraph at the end of the introduction (line 106-114 on the revised manuscript). This is to clarify the aim of the work is to relate smell loss recovery from COVID-19 to patient factors, as you identify, and we have clarified how by meeting this aim the work will be of use to readers e.g. clinicians who are looking for information to give to their patients, or researchers looking to conduct similar studies of their own.  

2) the abstract is confusing, please rewrite the entire paragraph.

The abstract has been re-written for clarity as requested. Perhaps the original version was trying to be overly brief, so the new version has been expanded, but is still in line with the word limit of 300 words.

3) the introduction is very short, several studies have been reported analyzing the oral manifestations of COVID-19, mention and comment on them ( see and discuss doi:10.3390/jcm9103218), it is necessary to stress how chemosensory impairments have been established as a specific indicator of COVID19.

We have considerably expanded the introduction, adding in several new references. Changes to the introduction include a section from lines 62-68 going into more detail on the theories by which COVID-19 can enter the oral tissues and damage sensory cells, referring to the suggested paper and others. There is also a large section from lines 77-104 on the revised manuscript referring to the reasoning for choosing the variables we chose to study and how these relates to the wider literature in terms of links to COVID-19 and/or links to impaired smell. This section also refers to the study you have linked, highlighting the importance of tooth loss due to periodontal disease, which was observed in 70% of patients in that study.

4) What are the data related to the oral condition of the patient? the sentence "The present study did not examine periodontal disease directly, however flossing may be associated with greater gingival health and reduced inflammation" is not consistent, please introduced other data to discuss and analyze previous findings regarding the importance of periodontal disease and how periodontal bacteria affect the oral cavity of the patient (please, see and consider: doi:10.3390/jcm9010284)

This has now been addressed by expanding on the discussion (lines 234-249). We have clarified that flossing has been shown to reduce periodontal inflammation (with references). We have also related the presence of inflamed periodontal pockets to a mechanism of entry of virus as suggested in the first paper you linked. We have also referred to periodontal pathogenic bacteria and how these are known to relate to systemic inflammatory disease, and made reference to the second suggested paper and how mouthwashes may be useful in COVID-19.

Minor issues:

  • Conclusions cannot be reduced to a list of sentences: you must improve them highlighting the limits and the future insights pointed out from this article.

We have expanded the conclusions (lines 281-293) to put the whole work into context with the wider literature and make some suggestions about future work, and further address the limitations of the present work.

  • The references must be reformatted according to the instructions for the authors (see Journal Articles: Author 1, A.B.; Author 2, C.D. Title of the article. Abbreviated Journal NameYearVolume, page range.)

We have reformatted all references in this way.

According to this Reviewer’s consideration, novelty and quality of the paper, publication of the present manuscript is recommended after major revision.

Thank you again for your comments,

Sincerely,

Dr A. Gardner

Ms. G. Catton

Reviewer 2 Report

The paper is interesting and methodologically correct. I do not have any particular suggestions for the authors. I would only suggest a more precise explanation of the usefulness of their retrieval with associated future directions.

The research deals with the assessment of smell recovery after COVID-19 and its association with oral factors. The research is novel and addresses an intriguing topic, often neglected in the hygiene-related practice. With respect to the published material, the present work addresses this specific topic in an unprecedented manner, dealing it with COVID-19, a complete novelty regarding smell and oral health, with novel results. Overall, the methodology adopted is correct and no specific faults have been found. Conclusions are consistent with the evidence and address the main questions of the paper, although a more precise characterization of the usefulness of authors'retrievals with associated future directions is needed. References, tables and figures are correct.

Author Response

Dear Colleague,

Thank you for taking the time to review our manuscript. We appreciate your comment on the need for clarity on the future directions. We have expanded the original manuscript both in the aims section and the conclusion to highlight our desired meaning regarding future directions. In the aims (lines 111-114 of the revised draft) we clarify that we are primarily hoping our work can be used to guide future studies regarding sample size and estimation of effect size, something notoriously difficult in the absence of prior literature. In the conclusions we reiterate this point (lines 187-294) and laso make suggestions about how future work could focus on targeted recruitment of patients affected by prolonged smell loss which may be useful to further understand the mechanism of COVID-19 mediated smell loss e.g. by analysing their oral epithelial expression of ACE-2 or their nasal microbiomes.

Thank you again for your comments,

Sincerely,

Dr A. Gardner

Ms. G. Catton

The paper is interesting and methodologically correct. I do not have any particular suggestions for the authors. I would only suggest a more precise explanation of the usefulness of their retrieval with associated future directions.

The research deals with the assessment of smell recovery after COVID-19 and its association with oral factors. The research is novel and addresses an intriguing topic, often neglected in the hygiene-related practice. With respect to the published material, the present work addresses this specific topic in an unprecedented manner, dealing it with COVID-19, a complete novelty regarding smell and oral health, with novel results. Overall, the methodology adopted is correct and no specific faults have been found. Conclusions are consistent with the evidence and address the main questions of the paper, although a more precise characterization of the usefulness of authors'retrievals with associated future directions is needed. References, tables and figures are correct.

Reviewer 3 Report

Interesting work with implications for a dentist daily practice..

How do authors explain the association of smell loss persistance with the use of floss? Or maybe it is just a coincidence?

We noticed several variables like "missing teeth, appliances or brushing frequency" were also statistically tested. What was the clinical substract for doing that?

Author Response

Dear Colleague,

Thank you for taking the time to review and provide comments on our manuscript. Our revised manuscript has been significantly expanded. Please find our responses to your individual points below.

Interesting work with implications for a dentist daily practice..

How do authors explain the association of smell loss persistance with the use of floss? Or maybe it is just a coincidence?

We agree there is a small chance (about one percent, based on the p-value of 0.01) of a coincidental relationship. What could be more likely is an inadvertent measure of some other health behavious. Alongside our re-iteration of not interpreting this finding causally we have expanded on this point in the relevant discussion section with some examples: 

“For example, flossing may be an inadvertent indicator of a generally more health-focused individual or perhaps someone more compliant and receptive to healthcare advice. Despite a statistically significant difference in flossers between smell recovery groups (75% of rapid smell recovery being vs. 58% of prolonged smell recovery) the clinical significance cannot be automatically assumed.”

Lines 271-275 of the revised manuscript.

We noticed several variables like "missing teeth, appliances or brushing frequency" were also statistically tested. What was the clinical substract for doing that?

We have expanded out introduction considerably to provide evidence-based rationale for our choice of all the variables we analysed, lines 77-104 on the revised manuscript. We expand on how brushing frequency or use of appliances such as dentures can be associated with changes in oral microbiome, which in turn may relate to sense of smell and the nasopharyngeal microbiome via retro-nasal olfaction. We also comment upon the previous finding of Sinjari et al., (ref 11.) which links tooth loss due to periodontal disease in COVID-19 infection, based on the recommendation of another reviewer. We thought tooth loss (not due to trauma or orthodontic reasons) could be indicative of general oral health, however it turned out within our sample the majority of individuals did not report significant tooth loss, possibly due to a slightly younger cohort on average.

Round 2

Reviewer 1 Report

The authors adequately addressed the recommendations of this reviewer